# Development of a Low-Cost Open-Source Measurement System for Joint Angle Estimation

**DOI:** 10.3390/s21196477

**Published:** 2021-09-28

**Authors:** Túlio Fernandes de Almeida, Edgard Morya, Abner Cardoso Rodrigues, André Felipe Oliveira de Azevedo Dantas

**Affiliations:** Graduate Program in Neuroengineering, Edmond and Lily Safra International Institute of Neuroscience, Santos Dumont Institute, Macaíba 59280-000, Brazil; edgard.morya@isd.org.br (E.M.); abner.neto@isd.org.br (A.C.R.); andre.dantas@isd.org.br (A.F.O.d.A.D.)

**Keywords:** sensors in healthcare, accelerometers, sensor fusion

## Abstract

The use of inertial measurement units (IMUs) is a low-cost alternative for measuring joint angles. This study aims to present a low-cost open-source measurement system for joint angle estimation. The system is modular and has hardware and software. The hardware was developed using a low-cost IMU and microcontroller. The IMU data analysis software was developed in Python and has three fusion filters: Complementary Filter, Kalman Filter, and Madgwick Filter. Three experiments were performed for the proof of concept of the system. First, we evaluated the knee joint of Lokomat, with a predefined average range of motion (ROM) of 60${}^{\circ }$. In the second, we evaluated our system in a real scenario, evaluating the knee of a healthy adult individual during gait. In the third experiment, we evaluated the software using data from gold standard devices, comparing the results of our software with Ground Truth. In the evaluation of the Lokomat, our system achieved an average ROM of 58.28${}^{\circ }$, and during evaluation in a real scenario it achieved an average ROM of 44.62${}^{\circ }$. In comparing our software with Ground Truth, we achieved a root-mean-square error of 0.04 and a mean average percentage error of 2.95%. These results encourage the use of this system in other scenarios.

## 1. Introduction

Movement analysis uses technological devices to identify problems or increase physical performance [1]. However, the gold standard of data acquisition systems used to measure movement is expensive and often requires highly controlled environments. A low-cost approach to replace the costly optical data acquisition systems for three-dimensional motion analysis is inertial measurement units (IMUs). A typical IMU comprises three sensors—accelerometer, gyroscope, and magnetometer—to measure linear acceleration, angular velocity, and local magnetic field. Recently, the combination of these measures has been introduced in rehabilitation science to perform an accurate assessment outside the laboratory [2].

The most common approach of combining IMU data is a fusion algorithm, specifically for measuring joint angles. For this, it is necessary to estimate the orientation of each IMU using sensor fusion filter algorithms [3]. Then, the gyroscope is used as primary reference due to the good response in high frequency; the accelerometer and magnetometer, which have a good response in low frequency, are used to correct the gyroscope drift [4,5,6]. After finding the IMU orientation, we can infer the joint angle through different solutions [7].

IMU has several applications for movement sciences, but its applied use must overcome some limitations. As suggested by [7], complex systems to estimate joint angles makes it hard to use in daily clinical practice. In addition, there exists a need to effectively combine accelerometer, gyroscope, and magnetometer data. A comprehensive literature review of IMU and methods used for motion analysis is presented. Seel et al. and Weygers et al. present a literature review on the use of IMU and methods used for motion analysis [3,7]. In general, research that presents systems for measuring angles does not provide the analysis codes, circuit connections, and algorithms used [7]. The system presented here aims to overcome this limitation, providing an easy-to-use, low-cost system for joint angle estimation.

This study aimed to (1) develop a low-cost, easy-to-build hardware to estimate angles, (2) present an open-source software to process IMU data with documentation, and (3) perform a system proof-of-concept by three experiments with available analysis and code.

## 2. Materials and Methods

In this section, there is material description, components, concepts, and techniques to develop the proposed system. Our hardware is named Joint Angle Measurement and Acquisition device (JAMA) and is composed of an IMU (GY-80) and a microcontroller (ESP32). The software library, fully developed in Python, was named Python for Joint Angle Measurement and Acquisition (PyJama), which implements three fusion filters for data processing. The system is available on GitHub (github.com/tuliofalmeida/jama (accessed on 1 July 2021) and github.com/tuliofalmeida/pyjama (accessed on 1 July 2021)) and was developed to work independently to analyze data collected with another IMU-based system or to use JAMA to collect data and analyze it with another software.

**Notation** **1.**
*γ denotes a general variable. $\vec{\gamma }$ represents a vector, $\hat{\gamma }$ represents an estimate. $\gamma^{*}$ represents the conjugated of γ. $\gamma_{\left[n\times m\right]}$ stand for a matrix with n lines by m columns. $\gamma^{T}$ represents the transposed of γ. ▽γ represents the optimizer for γ (here we use Gradient Descendant and Gauss-Newton). γ⊗γ represents that the vectorial product. $\dot{\gamma }$ represents the derivative of γ. γ·γ represents a scalar product. $∥\gamma ∥$ represents the normalized variable, $\bar{\gamma }$ represents the filtered variable. Δt is the sample time. $\gamma_{t}$ represents the variable in instant t and $\gamma_{t-1}$ represents the variable in instant t−1.*


### 2.1. JAMA

In this work, two JAMAs estimated knee joint angles. Each JAMA used one microcontroller ESP32-DevKitC (ESP32), one IMU GY-80 with 10 degrees of freedom, two batteries (3.7 V and 3800 mAh), one power button, and two input connectors to charge the batteries. The main characteristics of these components are low energy consumption, robustness, and low cost. Communication between ESP32 and GY-80 used an Inter-Integrated Circuit (I${}^{2}$C) [8] with an acquisition frequency of 100 KHz.

Each JAMA costs around 24 dollars (see Table 1) and has three main advantages: hardware size, Wi-Fi data transmission, and mobility (weighed 134 g, heavier than the gold standard devices, approximately 16 g).

#### 2.1.1. Data Acquisition

ESP32 extracts IMU data using I${}^{2}$C protocol with GPIO’S 21 and 22 connected to Serial Data (SDA) and Serial Clock (SCL) pins of the GY-80. The transmission starts when the SCL line becomes low. Using this logic, ESP32 controls the start and end of data sending in four steps: (1) signal generation to start transmission, (2) address transfer from GY-80, (3) data transfer, and (4) signal generation to end transmission [8].

Extracted data were organized in a buffer with the following sequence: accelerometer (*a*), gyroscope (ω), and magnetometer (*m*). Once the device was connected to Wi-Fi, a server was created at port 4000. These data were transmitted asynchronously over Wi-Fi (without any kind of processing), using the asynchronous Transmission Control Protocol communication (async-TCP) to respond to client requests (more details in Section 2.1.2). Additionally, the JAMA raw data service can be used by different analysis software, and all connections for building the circuit are presented in Figure 1. To wrap the circuit, we developed a 3D printing case using acrylonitrile butadiene styrene filament.

#### 2.1.2. Data Transmission

Data transmission based on a 2.4 GHz Wi-Fi network (without internet connection and with a range of 10 m) was designed to provide movement freedom out of the laboratory. This communication is an Asynchronous Transmission Control Protocol (async-TCP), used to prevent data from being lost during transmission, as it is performing in parallel with other microcontroller tasks. The async-TCP is a third-party library that enables a trouble-free, multi-connection network environment for microcontrollers. This library allows inertial data, collected by a microcontroller timer interrupt routine, to be written to a buffer and sent via TCP without interfering with data transmission. Moreover, if the data amount is large, the library manages the data transmission in such a way as to continue sending even after the data collection routine has finished.

Following the concept of asynchronous communication, the developed architecture is client/server-based. The client ran on an Acer Nitro 5 notebook (8th generation i5 processor with 16 GB RAM) connecting to the server via a hotspot (inherent in the Windows 10 operating system). The hotspot windows function creates a Wi-Fi network with an Internet Protocol (IP) address, network SSID, and password. The client application using Python (version 3.83) and communication with the ESP32 (distributed servers) connected to its IP. With the communication established, the ESP32 can send data to the client and stores the data in a text file.

Before starting the client API execution, it is necessary to edit the code with each JAMA IP address, the desired sample rate (pre-determined as 75 Hz, as suggested in the literature for gait task [9]), the experiment duration, the desired name for the text file, and the storage location. Our configuration allows the client to receive ten variables per pack in a sample time from each server (ESP32), namely the timestamp and the 3-axis accelerometer, gyroscope, and magnetometer.

The async-TCP uses two programming languages, C++ (server) and Python (client). Note that it is possible to configure the server directly on the hardware of each JAMA using C++ language. In code, one can set the network SSID and password so the JAMA can connect to the hotspot. After that, one can use the IP determined to identify each JAMA. Six steps summarize the JAMA operation:Build JAMA and check connections;Compile and upload firmware to each JAMA device (set IP, SSID and password);Activate the hotspot and check if all variables were correctly determined;Turn on JAMA devices, check the Wi-Fi connection, and place it in the joint;Run Python script to check connections and data streaming and recording;Run Python script to perform the data acquisition.

### 2.2. PyJama

PyJama is a Python library that processes data from JAMA or other devices (based on IMUs or optical systems).

Data pre-processing checks data integrity and sensor calibration, removes artifacts, and filters signals. Data processing combines accelerometer, gyroscope and magnetometer data to estimate IMU orientation and, therefore, joint angle.

Figure 2 shows PyJama classes with implemented functions. Requirements to use PyJama are numpy (v.1.20.0), SciPy (v.1.7.0), pandas (v.1.2.4), and matplotlib (v.3.4.2). For the development, we used data extracted from JAMA and gold standard devices (Total Capture (https://cvssp.org/data/totalcapture (accessed on 14 April 2021))) dataset [10].

#### 2.2.1. Data Handle

Data manipulation reads an organized data file to allow data processing functions. For this, *DataHandler()* class provides functions to read raw data generated by JAMA, Vicon Blade, and Xsens, and MTw wireless IMU organizes a dictionary or matrix (easily adapted to DataFrame, a spreadsheet with the data in columns), calibrates, and saves a csv file.

#### 2.2.2. Data Processing

*DataProcessing()* implements data processing and sensor orientation-related functions to manipulate Euler angles, quaternions, rotation matrix, and the conversion functions among them (Euler ⟷ quaternion ⟷ rotation matrix and Euler ⟷ rotation matrix). Additionally, this class implements the Gradient Descent and Gauss–Newton algorithms (Appendix A), sensor fusion algorithms for device data, low-pass filters for noise removal from raw signals, and simple joint angle calculation applied to each plane of motion.

PyJama calibrates accelerometers and gyroscopes by removing the average of the first 5 s of filtered static measurement from the entire signal. A $360^{\circ }$ rotation of the JAMA around the three axes of movement calibrates the magnetometers. This method is regularly used in magnetometers by rotating the sensors in the three axes one at a time [11].

PyJama implements fusion filters in quaternions. The motivation for using quaternions is the data stability, considering that the rotation using Euler angles can present more errors and the occurrence of Gimbal Lock [12,13]. Thus, Complementary Filter(CF) with Gradient Descent and Gauss–Newton [6,14], Kalman Filter (KF) with Gradient Descent and Gauss–Newton [6,14], and MadgwickAHRS [5] were implemented.

The complementary filter uses gyroscope measurements and complements the accelerometer and magnetometer measurements. Due to the behavior at low frequencies, the complementary filter uses the gyroscope data as a reference because the data present fewer variations, and an accelerometer and magnetometer are used to compensate for these low-frequency variations [4,15]. To handle data fusion, one can tune α, a weight to balance between gyroscope, and accelerometer and magnetometer data, which avoids neglecting the contribution of any measurement (accelerometer and magnetometer). In Algorithm 1, a Complementary Filter operation is implemented in PyJama. This allows the selection of one of the Gradient Descent or Gauss–Newton optimizers.

Kalman Filter (KF) aims to estimate linear dynamic system parameters (or states) that approximate measurements performed [16]. Many applications and possibilities of variations (Extended, Fuzzy, Unscented, and others) have made KF one of the most used filters to estimate variables indirectly. KF is a set of mathematical equations to estimate process states by minimizing the mean squared error [17]. Therefore, KF performs well in several aspects: it supports estimates of past, present, and even future system states, even when the precise nature of the modeled system is unknown. In an application using IMU data, KF predicts the orientation state using gyroscope data as a reference, similarly to complementary filter. The update step uses magnetometer and accelerometer data [6,14]. Additionally, the variable β determines balance among gyroscope, accelerometer, and magnetometer data. Moreover, KF can work in a real-time loop to be capable of predicting and correcting the IMU orientation [17]. The implemented KF algorithm is presented in Algorithm 2.

Similar to other presented filters, the Madgiwck filter uses gyroscope measurements as reference because of better behavior at high frequencies and interference isolation from forces during movement. Accelerometer and magnetometer measurements are used as precise instructions to compensate for gyroscope deviation in low frequencies due to the accumulation of numerical errors and influence of artifacts [5].

Formulations in quaternions require Madgiwck filter changes and specific implementation using the Gradient Descendant, as performed in its original formulation [5]. Madgwick filter implementation is presented in Algorithm 3.

**Algorithm 1** Complementary Filter

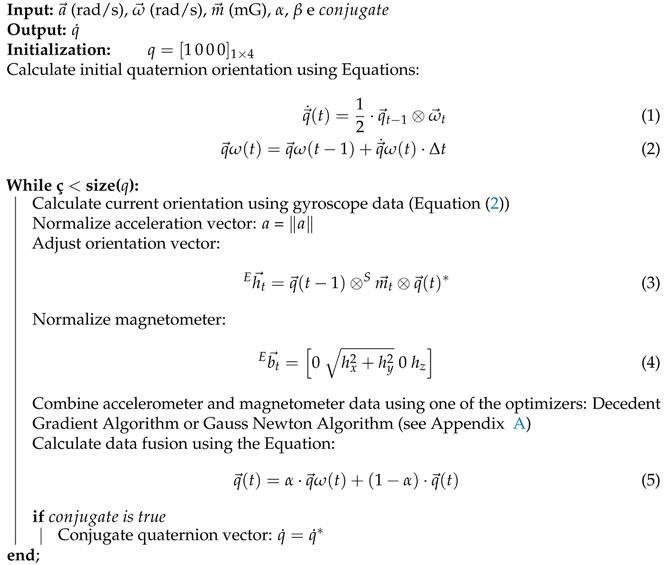



#### 2.2.3. Pattern Extraction

In particular, the human body has some cyclical movements, such as gait, and these cycles are traditionally identified and extracted by kinematic analysis [2,18,19]. In this sense, a function developed to elicit these cyclic patterns is presented in Algorithm 4. In addition to pattern evaluation, this function implements statistical metrics extracted from patterns found, returning the confidence interval, standard deviation, polynomial coefficient, coefficient of determination, and variance. Specifically, the polynomial adjust returns minimum and maximum value estimates (calculated using percentage of gate cycle and coefficients output). We also have the *data_comparison()* function, which plots the polynomial fit curves of two measures for visual comparison of the data.

#### 2.2.4. Data Visualization

An essential part of data analysis is its visual representation with figures and graphs. The *DataVisualisation()* class implements automatized plots in single figures or sub-figures, identifies the size of the input data, and organizes the entire figure layout. The standardization of the axes is automatic, and it is simple to determine captions, titles, and subtitles. Structurally, the class functions use the matplotlib. In addition, users can choose to use and visualize the data with the orientation in quaternions or Euler angles, the last is the most common way to visualize joint rotations [12].

#### 2.2.5. Data Analysis

The *DataAnalysis()* class is easy to use for users with little data analysis experience. This class has functions that perform data processing in a few steps. It has functions to apply all implemented filters, perform all metric calculations and pattern extraction, and estimate the angle of a joint. In other words, it has a complete analysis routine implemented in these functions.

**Algorithm 2** Kalman Filter

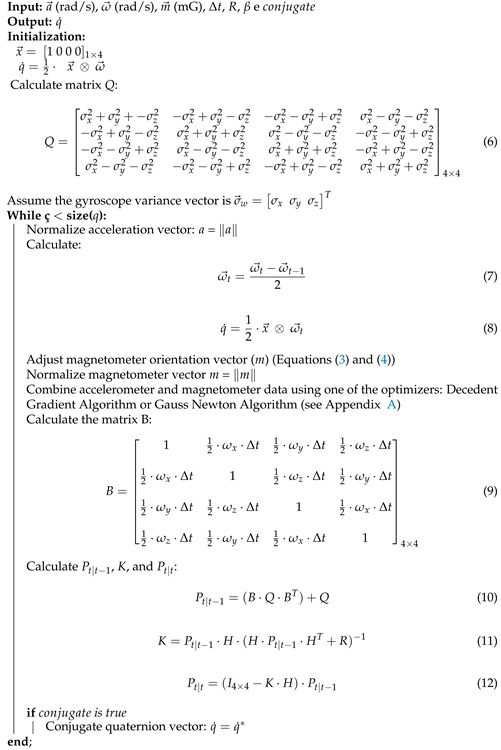



**Algorithm 3** Madgwick Filter

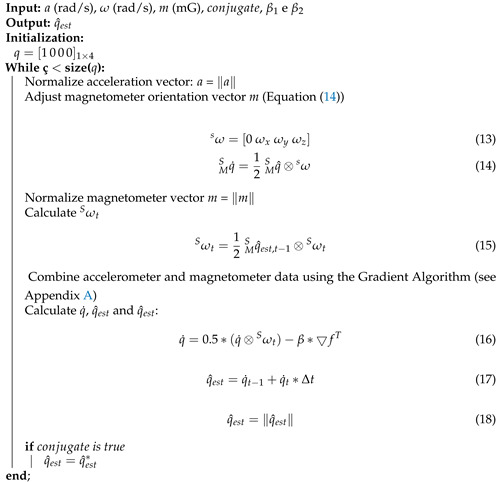



### 2.3. Experiments

In this section, the methodology used to prove the concept of the developed system is presented in three experiments. JAMA was attached to rehabilitation robot (Lokomat) and analyzed the data using PyJama. Second, JAMA was attached to the knee joint of a walking person on a treadmill (again using JAMA with PyJama). Third, PyJama processed an open dataset with gold standard device data.

This project was approved by the ethics committee of Federal University of Rio Grande do Norte under the protocol 79649717.0.0000.5292 approved in 15 December 2017.

#### 2.3.1. Experiment 1

A controlled task allowed to assess JAMA measurements and joint angle estimation using PyJama. JAMA was attached to hip and knee joints of Lokomat (HocomaAG, Volketswil, Switzerland), a validated robot for gait rehabilitation with sagittal plane movements [20]. The recording of the experiment was to compare the measurement obtained by both JAMA and Kinovea, a free and validated software to assess joint angle in 2-dimensional videos [21].

Lokomat is a market-leading rehabilitation robot capable of providing highly repetitive gait training with physiological movement patterns. It consists of a pair of robotic legs and a user’s weight support device. The operator can determine joint angles and simulate gait speed [20], a controlled scenario to test JAMA measurements. For this test, the range of motion was 45${}^{\circ }$ for the hip joint and 60${}^{\circ }$ for the knee joint, and walking speed was 1 km/h. The joint evaluated was the knee, with two JAMA devices attached to the sides of the right robotic leg (with the *Y*-axis aligned with gravity), acquisition frequency of 75 Hz, and 120 s of data transmission.
**Algorithm 4** Pattern Extraction
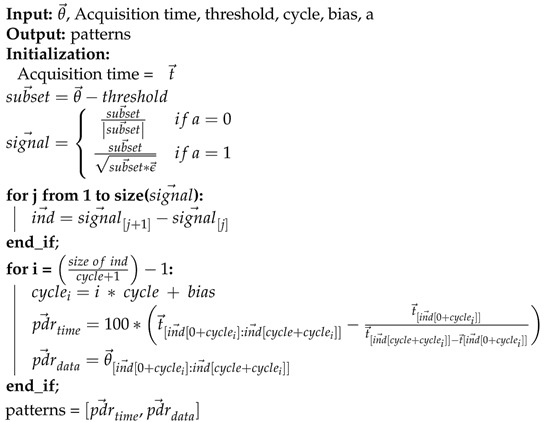


Video recording used an iPhone 8 (Apple Inc., Sunnyvale, CA, USA) with a resolution of 1920 × 1080 at 30 frames per second. JAMA was attached to the Lokomat using double-sided tape (3M Company, Saint Paul, MN, USA). Kinovea software tracked two points frame by frame: one in the hip joint and the other in the distal part of the robotic leg. Kinovea exports a .csv file with the angles frame by frame. Figure 3 presents the experimental configurations and parameters.

#### 2.3.2. Experiment 2

Experiment 2 evaluates JAMA in a human gait task on a Lokomat treadmill with a speed of 4 km/h, as proposed by Lebleu et al. [2] to evaluate IMU sensors in an adult male, 184 cm tall, body mass index of 20.7 kg/m${}^{2}$, with no gait disability or previous injuries in the last 6 months. Two devices were positioned, on the thigh and on the shin [1,2,3], with sampling rate of 75 Hz and 120 s of acquisition time. The experiment setup is shown in Figure 4. Regarding video recording, the assessment was realized under the same conditions as Experiment 1 and was also analyzed using Kinovea.

#### 2.3.3. Experiment 3

Performance of the PyJama was evaluated with data from gold standard devices available in the Total Capture dataset [10]. Data were collected using the optical device Vicon Blade and the inertial sensor Xsens MTw wireless IMU with an acquisition frequency of 60 Hz.

Data were collected indoors, using eight high-definition Vicon Blade cameras (with 60 frames per second) duly calibrated with 2.5-m supports and 13 Xsens sensors distributed in a capture suit. The dataset provides the quaternion orientation from the Vicon system; the accelerometer, gyroscope, and magnetometer provided raw data, and quaternion orientation (Ground Truth) was taken from Xsens sensors. A walking task with markers and sensors on the thigh were used to test PyJama.

A comparison performed between orientations provided in quaternions by Vicon and Xsens showed a significant difference. Thus, Vicon’s data to compare with PyJama’s estimates were discarded. Probably, this difference was due to the positioning of Xsens sensors and Vicon’s body markers, which require some specific type of adjustment. The calibration done by the authors considered the orientation of the bone of a virtual avatar [10], which was not directly related to the context of our work. Accelerometer, gyroscope, and magnetometer data to estimate the orientation using the filter algorithms implemented in PyJama were compared to Ground Truth. Comparisons were made using root-mean-square error (RMSE) and the mean absolute percentage error (MAPE) metrics. RMSE is the most common metric in studies with sensors [7], and MAPE is an error metric that is easier to be interpreted. For this experiment, joint angles were not calculated since the dataset does not provide it.

## 3. Results

This section presents the results of experiments performed to proof the concept for both JAMA and PyJama. All results are interactively available on Google Collaboratory notebooks, hosted on PyJama GitHub. In addition, a step-by-step tutorial on how to build and use JAMA is available on GitHub.

### 3.1. Experiment 1

In Table 2, calculated metrics based on the gait cycle extracted within the function *joint_measures()* from *DataAnalysis()* class were presented.

In this experiment the minimum point was 0${}^{\circ }$, which does not affect the analysis and explains the estimate values being a little different from those found. According to the data presented in Table 2, the developed filters were able to estimate the expected ROM of 60${}^{\circ }$. Despite similar results, the Madgwick Filter (58.28 ± 2.75${}^{\circ }$, CI = 5.39%) showed the best results, followed by the Kalman GD Filter (58.81 ± 2.90${}^{\circ }$, CI = 5.68%), Kalman GN Filter (56.81 ± 2.81${}^{\circ }$, CI = 5.51%), GD Complementary Filter (60.89 ± 4.81${}^{\circ }$, CI = 9.43 %), and GN Complementary Filter (61.57 ± 4.94${}^{\circ }$, CI = 9.69%), this considering the averages, standard deviation, and confidence interval.

JAMA and Kinovea data were compared using the gait pattern average cycles of Lokomat and extracted by PyJama. Due to its better performance, Madgwick Filter was used to find the cycles. To facilitate visual comparison, we calculated the cycles for Madgwick Filter and Kinovea, which are presented in Figure 5 using the *data_comparison()* function.

As expected, the results obtained with the application of the fusion filters were consistent, and Figure 5 shows the expected gait pattern with the dispersion metrics found (Table 2). Note that parameters used to tune filters were generalist. In other words, a specific adjustment of parameters of each filter might present better results.

### 3.2. Experiment 2

Experiment 2 evaluated the gait of an adult participant using Madgwick filter, and the results in Experiment 1 support its use to analyze gait data.

Table 3 presents values for knee joint using Madgiwck’s Filter. Measurement variabilities of Kinovea (dp = 5.9${}^{\circ }$, var = 35.47${}^{\circ }$) were similar to PyJama. Due to standard human gait variability, a target ROM was not determined. Additionally, there was no difference between measured and estimated Min/Max, possibly attributed to a variance effect, although Min/Max of the overall data is subject to uncertainties. However, estimated Min/Max from a polynomial curve based on least-squares data removes Gaussian noise.

Similar to Experiment 1, a threshold determined subsets of gait cycles, and a similar joint behavior (flexion and extension) was extracted from JAMA and Kinovea.

Figure 6 shows a comparison between knee flexion and extension movement average with similar patterns of Madgiwck and Kinovea. The knee joint had a well-known pattern in the gait cycle, and both methods presented two flexion peaks. However, extracted cycles had inverted gait phases, starting from the swing phase to stance phase. Because the function *pattern_extraction()* has no parameters to adjust the order of the phases, only the input threshold was presented in Algorithm 4. Other influencing factors might be gait speed (4 km/h), JAMA limb positioning, and a difference in acquisition frequency between JAMA and the video (Hz and fps).

### 3.3. Experiment 3

Experiment 3 evaluated the manipulations and transformations performed by PyJama on Total Capture [10] dataset. Total Capture dataset has Vicon data in quaternion orientation, and Xsens has raw and quaternion orientation data. Reading functions for Vicon and for Xsens data, respectively *vicon2dict()* and *xSens2dict()*, returned data in a dictionary form to ensure analysis.

Comparing quaternion data from Vicon and Xsens (RMSE and MAPE for each quaternion—q1, q2, q3, and q4—and quanternions average) obtained an average RMSE of 0.69 ± 0.04 (q${}_{1}$ = 0.67, q${}_{2}$ = 0.70, q${}_{3}$= 0.74, q${}_{4}$ = 0.63) and an average MAPE of 64.40 ± 11.60% (q${}_{1}$ = 53.72%, q${}_{2}$ = 57.71%, q${}_{3}$ = 79.94%, q${}_{4}$ = 66.21%).

Quaternion orientation data from the Xsens sensors were used as Ground Truth for manipulations, and estimates of fusion filters were implemented in PyJama as proof of concept. Estimates using accelerometer, gyroscope, and magnetometer raw data were extracted from algorithms implemented in the PyJama *DataProecessing()* class.

First, the manipulation test transformed Xsens reference data to Euler angles and transformed them back to quaternions (quaternion ⟶ Euler ⟶ quaternion). This manipulation presented an average RMSE of 0.73 ± 0.05 (q${}_{1}$ = 0.67, q${}_{2}$ = 0.73, q${}_{3}$ = 0.80, q${}_{4}$ = 0.70) and an average MAPE of 55.72 ± 6.57% (q${}_{1}$ = 47.95%, q${}_{2}$ = 55.35%, q${}_{3}$ = 64.03%, q${}_{4}$ = 55.56%). However, the function conversion from quaternions to Euler angles uses an in *pitch* rotation sine arc, which limits angle results to the range of $I=\left\{-\frac{\pi }{2}\leq \theta \leq \frac{\pi }{2}\right\}$, causing errors. Quaternion vector values were limited to two quadrants, using absolute values, and showed an average RMSE of 7.41 $\times 10^{-7}\pm$ 3.15 $\times 10^{-7}$ (q${}_{1}$ = 5.16 × $10^{-7}$, q${}_{2}$ = 5.94 × $10^{-7}$, q${}_{3}$= 1.28 × $10^{-6}$, q${}_{4}$ = 5.69 × $10^{-7}$) and an average MAPE of 3.69 × $10^{-5}$± 2.15 % (q${}_{1}$ = 3.13 × $10^{-5}$ %, q${}_{2}$ = 3.69 × $10^{-5}$ %, q${}_{3}$ = 7.72  × $10^{-5}$ %, q${}_{4}$ = 3.57 × $10^{-5}$ %) as presented in Figure 7.

Table 4 shows PyJama quaternion orientation estimation, using the Complementary Filters (with GD and GN), Kalman (with GD and GN), and Madgwick, all of them with the orientation constraint.

PyJama’s filters showed similar results with an average RMSE of 0.73 and a MAPE of 59.6%. Nevertheless, restriction of quadrant showed a better result again. Thus, Complementary Filter with GN (presented in Figure 8) had the best performance, which presented an average RMSE of 0.04 ± 0.01 and MAPE of 2.95 ± 0.51% followed by Madgwick (RMSE = 0.05 ± 0.01, MAPE = 4.52 ± 0.88%), Complementary Filter with GD (RMSE = 0.11 ± 0.02, MAPE = 8.40 ± 1.32%), Kalman with GN (RMSE = 0.11 ± 0.11, MAPE = 9.85 ± 1.22%), and Kalman with GD (RMSE = 0.11 ± 0.01, MAPE = 10.63 ± 1.18%). The results suggest that filters with GN performed better than those with GD. However, this might be related to a classic GD local minimum problem.

Note the measurement errors at the initial moment were caused by the filter adjustment delay. The first estimate is predetermined in the initial orientation input ([1,0,0,0]) and then updated with the input data. If this initial data are better adjusted, the results of all measurements tend to have a lower RMSE and MAPE.

## 4. Discussion

### 4.1. JAMA

This paper presented a low-cost device capable of extracting data from an IMU. Unlike other works [7,22,23], in addition to presenting a low-cost device, we provide how to build and use it. Although we have limitations, this availability facilitates integration with the community and the development of JAMA variations, as it is possible to change the batteries, the sensor, or even modify the case to improve the measurement performance.

The biggest limitation in using JAMA is the non-validation of two measurements. Most of the works that develop hardware using IMU use gold standard devices as a reference during development and for measurement validation [2,3,24]. Although it is not the best technique for device validation, the use of robots is a form of validation, and here we use Lokomat to carry out the proof of concept of the JAMA measurements [20,24,25,26]. In this context, the signals extracted by JAMA present an expected pattern, which can be observed based on the results of the performed experiments but also on the shape of the extracted signal (data presented on GitHub).

In this paper, for proof of concept, an acquisition frequency of 75 Hz were used, which can extract enough data for the algorithms developed with PyJama [9]. Some algorithms need a higher amount of data to work, around 200 Hz [27,28]; this is a limitation of JAMA, considering that the GY-80 continuous measurement mode operates at frequencies of 0.75–75 Hz, while the single measurement mode operates at frequencies of 0.75–160 Hz, but packet loss may occur.

JAMA may not present good results for evaluating rapid movements, such as running (>8 km/h), in its current version. On the other hand, these hardware configurations are compatible for gait assessments [22,29], assessment of falls in the elderly [30], and clinical tests (Time Up and Go and 6-min walk test) [31]. Therefore, based on the results presented, JAMA can perform IMU guidance data extraction.

### 4.2. PyJama

The main objective for the development of PyJama is to provide open-source algorithms for processing data from inertial sensors in different stages (pre-processing, processing, data analysis). There are several systems available for measuring angles, whether commercial or free, but few of them are open-source [7] and specially developed in Python. As Python is the most used programming language in science, PyJama arises to minimize this gap.

Fusion filters are one of the options for estimating orientation of inertial sensors. PyJama has the following Fusion Filters: Complementary Filter, Kalman Filter, and Madgwick Filter, and all of them use accelerometer, gyroscope, and magnetometer measurements to estimate orientation. By combining these different measurements, fusion filters provide more stable results for simpler sensor noise removal [5]. These filters have become the accepted basis for the majority of systems that use low-cost sensors and present the real-time feature [5,7,32]. Although our scope is to analyze offline data in Google Colab, PyJama has filters implemented for real-time use (update functions presented in Figure 2).

Commercial systems such as Xsens [27] and Physilog [28] use the strap-down integration technique to merge IMU data [7]. However, it is not indicated for sensors with a less robust measurement, and it is necessary to use a higher acquisition frequency [7,27,28]. Therefore, when using low-cost sensors for data acquisition, fusion filters present better results.

In the state-of-the-art time series analysis for pattern recognition, information extraction and data fusion have been made by artificial neural networks in different areas [33,34,35]. Recently, these techniques have also been applied for IMU data fusion. Weber, Gühmann, and Seel [36] developed RIANN (Robust Neural Network Outperforms Attitude Estimation Filters), which implements a Recurrent Neural Network with superior results to the fusion filters. However, even with the inferior results, the authors point out that the fusion filters are granted additional advantages of parameter optimization on target sequences and have a lower computational cost.

A similarity between filters implemented in PyJama and most neural network architectures is the use of optimizers. These optimization techniques aim to decrease the cost function, in other words, to decrease the measurements error [5,6]. PyJama implements filters using the most common optimization algorithms, Gradient Descent and Gauss–Newton. These algorithms allow quick correction of measurements, minimizing noise and sensor measurement errors.

For data drift removal, a sensor calibration based on static upright pose measurements was implemented for the gyroscope and accelerometer measurements. More complex and/or faster movements tend to encounter greater measurement errors; for these cases other techniques may be needed, such as functional calibration, as presented in [2].

### 4.3. Experiment 1

Experiment 1 aimed to evaluate the performance of JAMA and PyJama in a controlled situation of knee joint movement with a robotic leg (Lokomat). Overall, PyJama estimated the knee angle correctly, and all filters showed an average range of motion of 60${}^{\circ }$, with better performance of Madgwick filter (Table 2).

However, notwithstanding the metrics found are correct, the graphical representation (Figure 5) of the gait cycles was inverted (with swing phase, the peak of 60${}^{\circ }$, at the beginning of the cycle). This inversion was due to the pattern extraction algorithm developed (Algorithm 4). The inverted representation for the gait cycle is unusual, but some authors present their results in this way, even using commercial analysis systems [37], which shows that this is a crucial error in the algorithm.

The algorithm for pattern extraction presented was developed based on the angle and not the temporal perspective, which is the most common [2,3,37]. Thus, when using the angle as a more simplified way to extract the cycle information, the algorithm lost the perspective of the time series and inverted the graphical representation, but without changing the metrics of the angle estimated by PyJama and Kinvoea.

Kinovea has adequate precision for biomechanical analysis [21]. Therefore, we used its measurements to assess the behavior of fusion filters, even though it is not the gold standard. As a result, PyJama proved to be able to analyze data from other sources, and the implemented filters were able to estimate the angle with results closer to those found by Kinovea. Note that it was not the objective of the experiment to evaluate the individual performance of each filter. They were all tuned using the main input parameters. Therefore, when using specific parameter combinations for each filter, the results tend to improve [38]. Data and codes are available from the aforementioned GitHub library.

### 4.4. Experiment 2

Experiment 2 aimed to test JAMA and PyJama in a real-world scenario evaluating the knee joint of a healthy adult during gait. Data acquired using JAMA and analyzed using the PyJama Madgwick Filter resulted in an average ROM of 44.62 ($\pm 3.90^{\circ }$), and for Kinovea the average ROM was 50.93 ($\pm 5.96^{\circ }$).

Walking at a speed of 1 m/s or 3.6 km/h, the knee ROM ranges between 50 and 60${}^{\circ }$ [2,3,18]. Individual factors (such as height and size of the legs) affect the biomechanics of movement and, consequently, the kinematic metrics. In the case of this experiment, we expected an ROM of less than 60${}^{\circ }$ due to individual characteristics (Section 2.3.2). In addition to these factors, data may have suffered interference from the pattern extraction Algorithm 4, which caused a difference of 15 degrees during the swing phase between Kinovea and PyJama estimation.

In addition to the inversion of the phase representation, the algorithm proposed for pattern extraction was also not able to deal with the variability of human movement. An average ROM less than 60${}^{\circ }$ happened due to the influence of the algorithm errors. In other words, when extracting a wrong pattern, the algorithm changed the data mean. Table 3 presents the experiment metrics, where the minimum and maximum values of the measurements appear to be correct.

The difference between the estimate of PyJama and Kinovea was based on data behavior. Kinovea’s data showed less variability because they were extracted by video, while data extracted by JAMA suffered from interference from noise produced by the body itself, such as soft tissue noise [39]. Again, the filter was not optimally tuned to merge the data, which can improve the orientation estimate [38]. Based on other studies, gait speed was not a determining factor for the variability presented, considering that, at speeds in a range of 2 to 5 km/h, the knee joint presented an amplitude of 60${}^{\circ }$ [2,3,18].

The most used way to extract patterns from the cycles is based on kinematic metrics, such as the size and duration of a normal step [1,3,40]. However, many of these methods would not work out for our application because they are based on ground contact measurements. Alternatives of extraction are the use of machine learning to classify patterns. Gaitpy is a free python software that extracts kinematic gait parameters based on measurements from just an accelerometer positioned over the lumbar spine [29]. PyJAMA also implements this experiment, and the codes and data are available on GitHub.

### 4.5. Experiment 3

Experiment 3 aimed to validate fusion filters implemented in PyJama. As it was not possible to make comparisons between our system and gold standard devices for motion analysis, we chose to analyze data from these devices to validate PyJama. In this sense, we choose the Total Capture dataset [10] for providing synchronized data between two gold standard systems, using Xsens and Vicon. Other datasets involving sensors are available for free, but some have download problems, do not have the IMU and optical synchronized data attached, and/or do not provide raw and processed data. In this sense, the Total Capture dataset [10] was chosen for providing synchronized data between two gold standard systems, using Xsens and Vicon.

First, Xsens quaternion data were transformed to Euler angles, and then transformed back to quaternions. This operation presented an RMSE of 0.00 and MAPE of 0.00% (Figure 7), which demonstrates that the transformation functions implemented works correctly. After that, PyJama filters procesed Xsens data (accelerometer, gyroscope, and magnetometer) to estimate quaternion orientation and compare with Ground Truth.

The implemented filters showed good accuracy. The higher error without the quadrant restriction was something to be expected due to the different analysis techniques used in the formulation of the dataset [10,27]. However, after quadrant restriction, PyJama filters showed good accuracy (RMSE 0.04 and MAPE 2.95%, as the RMSE (in angles) for systems with high precision is around 1.3–11.22 degrees [7]; results are presented in Table 4). The RMSE is the most used error metric in studies with IMU, but here we also calculate the MAPE as its interpretation is easier. It will return the difference between the data regardless of the analyzed unit (degrees, quaternions, centimeters, and others).

We carried out an evaluation here for proof of concept of the filters. In this sense, we also standardized the input parameters (balance between accelerometer, gyroscope, and magnetometer contributions and input data filtering). It is not possible to have a fair performance comparison among filters, as they have not the best tuning [38]. Therefore, it is necessary to adjust parameters using specific algorithms to compare the performance of filters. The most used for this purpose is the genetic algorithm [41,42]. This algorithm belongs to the class of evolutionary algorithms. It aims to approximate the best possible solution using a random set of input parameters. After the initial estimate, you can evaluate the results and change the parameters at each cycle. This process is repeated until reaching the programmed limits (number of interactions or error metric).

Probably, the non-optimization of parameters negatively influenced the evaluation of Kalman filters with greater intensity and was probably for two reasons: (1) the implementation used was proposed by Comotti [6], where he performed the comparison between the Kalman filter and the Complementary Filter, with the Kalman Filter showing a better performance; and (2) due to gyroscope calibration. The Kalman filter function uses the first few seconds of the gyroscope data (static upright pose) to calibrate future measurements based on the variance of the data. As the Total Capture dataset was used, these data are not provided, which led to a longer adjustment time for the Kalman Filter. Therefore, these two factors contributed negatively to the Kalman filter measurements.

In general, the functions implemented in the PyJama were effective for processing the IMU data and fusion filters in real-time through the update functions (Figure 2).

### 4.6. Limitations and Perspectives

This study focused on a proof of concept, testing the systems with a rehabilitation robot and a healthy adult individual evaluating the knee joint in the sagittal plane. Tests for other joints and in other populations are needed in future studies, mainly for system validation.

We based the proposed pattern extraction on the angle (Algorithm 4). This approach proved to be inferior to other techniques proposed in the literature (for not being able to deal with the variability of movement), specifically due to the patterns extracted based on time [3,29]. Because of this, other algorithms for pattern extraction will be implemented in the future.

Although the system was able to extract the behavior of the joint, and specifically PyJama achieved a high accuracy when processing Xsens data, we can improve our system by adding algorithms to locate the joint centers and improve the robustness of the segment orientation automatically [43,44]. Another important factor is the GUI (graphical user interface) to guide user and facilitate its use in clinical applications.

## 5. Conclusions

This paper proposed a low-cost system for measuring joint angles in three experiments as a proof-of-concept of JAMA and PyJama. We used PyJAMA to process the data collected by JAMA in two experiments, which was able to find some pattern behavior of the knee joint. In the gold standard data processing tests, PyJama reconstructed the original signal (RMSE 0.00 and MAPE 0.00%) and estimated the sensor orientation with a low error (RMSE 0.04 and MAPE 2.95%). The proposed system needs some specific conditions for sensor calibration. In addition, it needs a long time for sending async-TCP buffer data. In future work, we plan to enhance the communication protocol and implement new functions for data calibration and fusion, using different techniques.

## Figures and Tables

**Figure 1 sensors-21-06477-f001:**
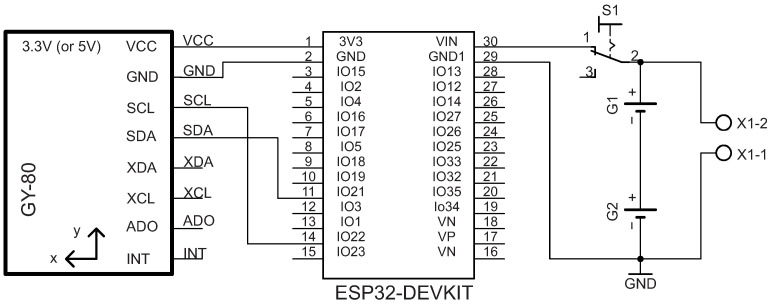
JAMA circuit can be configured to ESP32 to connect to GY-80 via I${}^{2}$C master mode. The microcontroller uses pins 1 and 2 for power supply (the I${}^{2}$C typically supports 3.3 V or 5 V), and pins 21 and 22 connect to the bidirectional transmission lines Serial Data (SDA) and Serial Clock (SCL), respectively.

**Figure 2 sensors-21-06477-f002:**
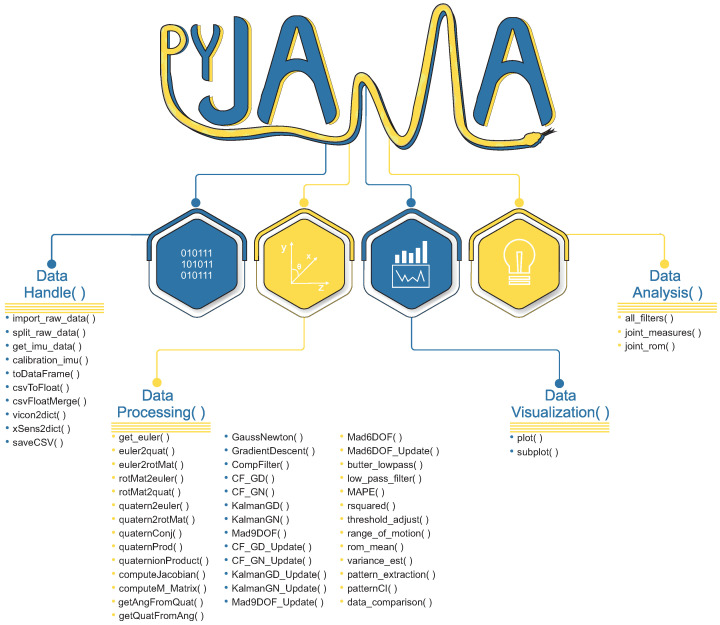
PyJama organization consists of four main classes to handle data formats, processing data, visualization, and analyze data.

**Figure 3 sensors-21-06477-f003:**
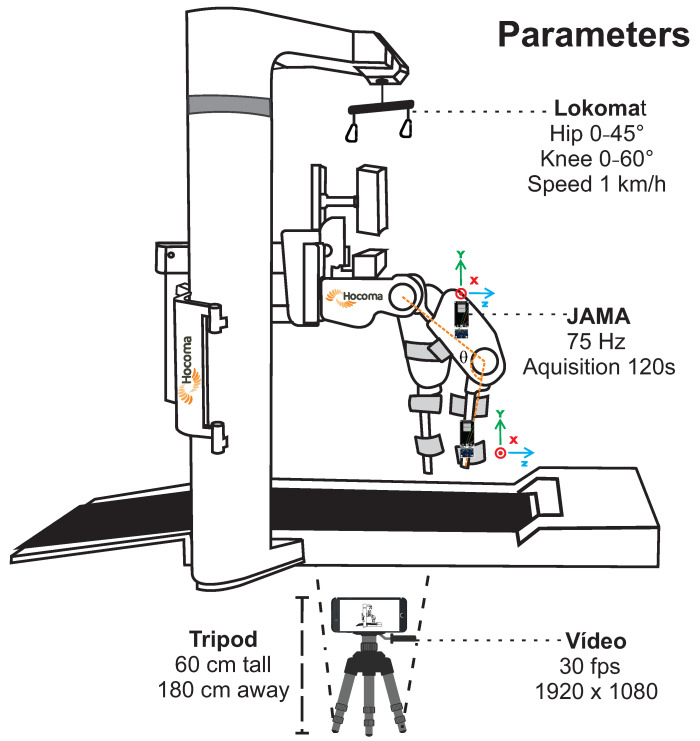
Experiment 1. Figure shows Lokomat schema (sidebars removed for better viewing) and parameters used for data acquisition. Two JAMAs were positioned on the side of the robotic leg, one above and the other below the knee joint, using 75 Hz as the acquisition frequency. Camera was positioned 180 cm away on a 60 cm height tripod. Video was recorded in full HD resolution at 30 fps. Lokomat was configured for a 45${}^{\circ }$ hip and 60${}^{\circ }$ knee ROM and simulating a 1 km/h gait.

**Figure 4 sensors-21-06477-f004:**
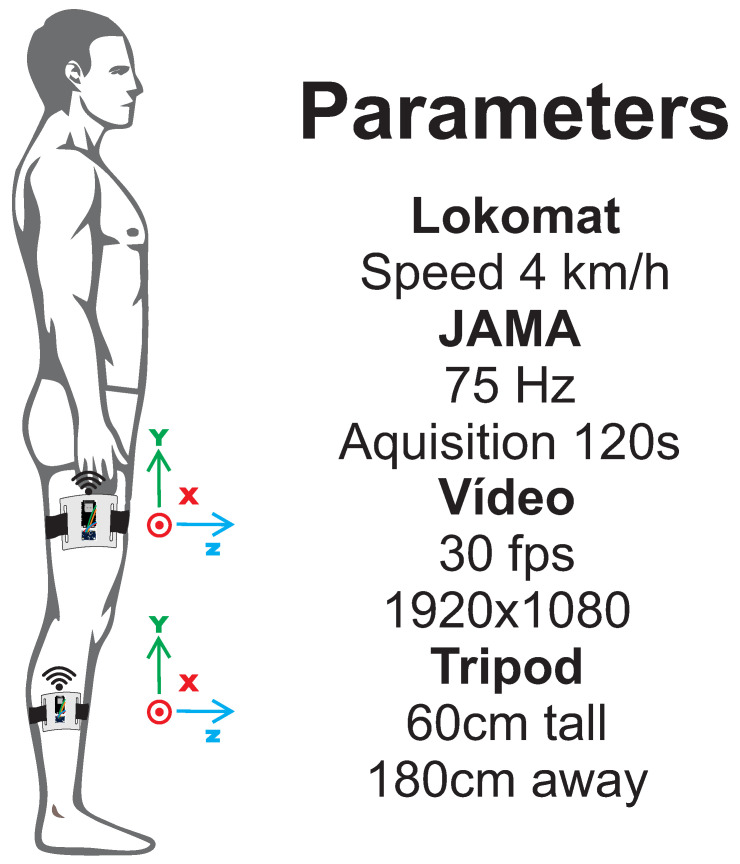
Experiment 2. Figure shows the positioning of the two JAMAs used to assess the knee joint with the parameters for data acquisition. One JAMA was positioned above and the other below the joint, using 75 Hz of acquisition frequency. Camera was positioned 180 cm away and 60 cm high on a tripod. Video was recorded in full HD resolution at 30 fps.

**Figure 5 sensors-21-06477-f005:**
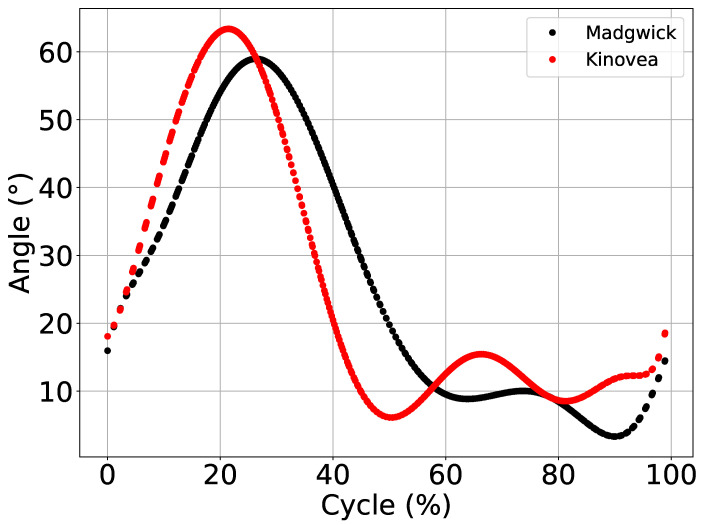
Data comparison between Madgwick and Kinovea. Figure shows the values of average ROM adjusted based on Min and Max, estimated based on the polynomial adjust.

**Figure 6 sensors-21-06477-f006:**
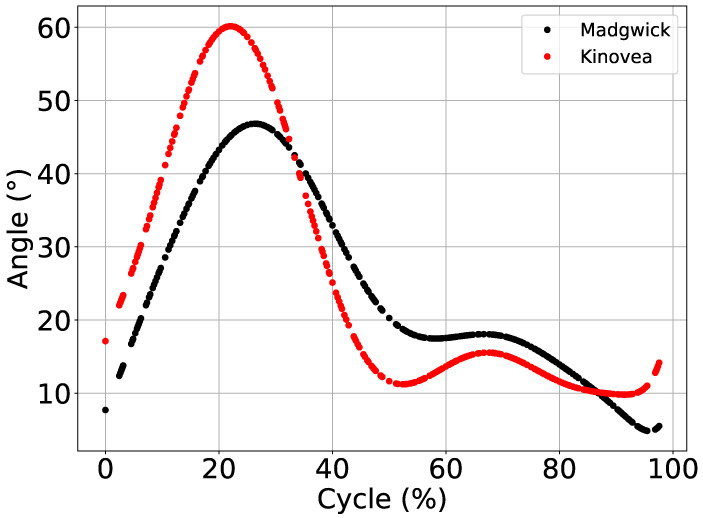
Comparison between Madgwick and Kinovea processed data. Figure shows the values of average ROM adjusted based on Min and Max estimates, based on the polynomial adjust.

**Figure 7 sensors-21-06477-f007:**
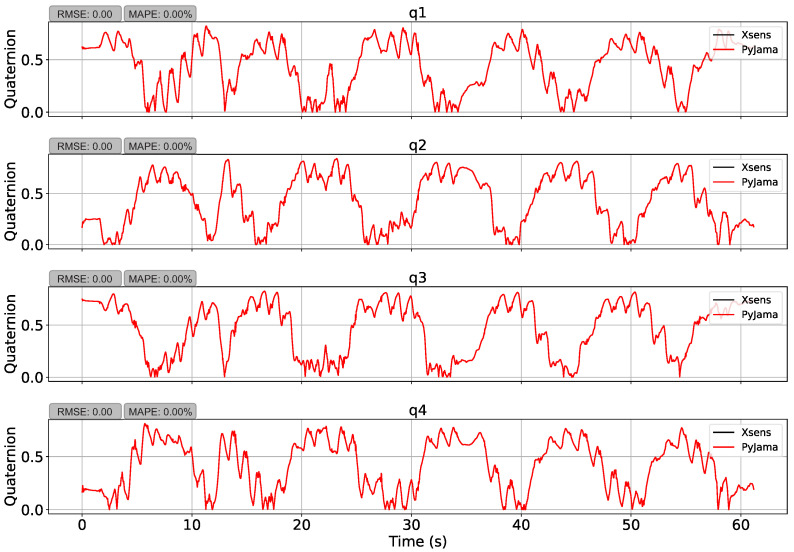
Data manipulation by PyJama. The figure shows the transformation of Xsens quaternion data to Euler angles, and from Euler angles back to quaternion, using quadrant constraint. The manipulation showed an RMSE of 0.00 and a MAPE of 0.00%.

**Figure 8 sensors-21-06477-f008:**
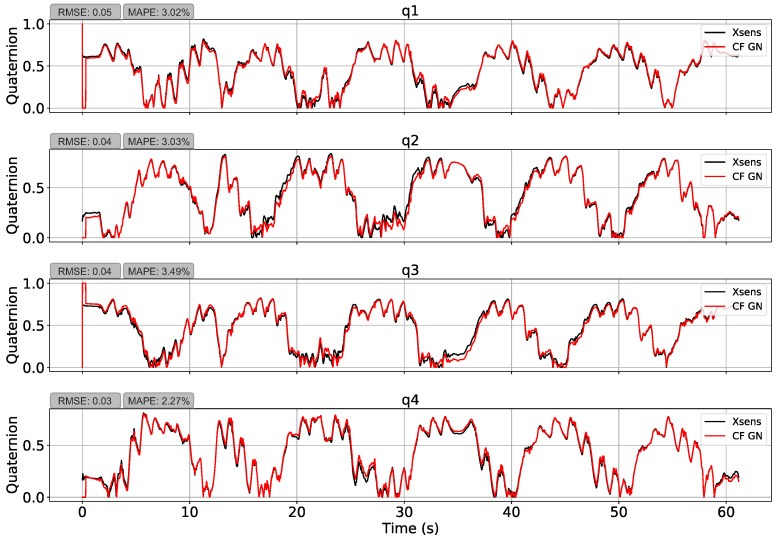
Orientation estimation using the Complementary Filter with Gauss–Newton. The estimation resulted in an average RMSE of 0.04 and an average MAPE of 2.95%.

**Table 1 sensors-21-06477-t001:** Hardware components.

Components	Price ($)	Quantity (Unit)	Total ($)
ESP32	5.00	2	10.00
GY-80	15.00	2	30.00
Batteries 3.7 V	4.00	4	8.00
Total	–//–	–//–	48.00

IMU can be customized according to user preference and price. For example, the MPU-9250 costs $5.00.

**Table 2 sensors-21-06477-t002:** Experiment 1—Metrics.

Movement/Filter	ROM Max	Target ROM	Average ROM (sd)
Flex/Ext CF GD	72.78${}^{\circ }$	60.00${}^{\circ }$	60.89 ± 4.81${}^{\circ }$
Flex/Ext CF GN	73.58${}^{\circ }$	60.00${}^{\circ }$	61.57 ± 4.94${}^{\circ }$
Flex/Ext Kalman GD	70.91${}^{\circ }$	60.00${}^{\circ }$	58.81 ± 2.90${}^{\circ }$
Flex/Ext Kalman GN	69.14${}^{\circ }$	60.00${}^{\circ }$	56.81 ± 2.81${}^{\circ }$
Flex/Ext Madgwick	65.92${}^{\circ }$	60.00${}^{\circ }$	58.28 ± 2.75${}^{\circ }$
Flex/Ext Kinovea	61.59${}^{\circ }$	60.00${}^{\circ }$	56.25 ± 1.94${}^{\circ }$
**Movement/Filter**	**CI 95%**	**Min/Max**	**Min/Max Est**
Flex/Ext CF GD	9.43%	0.00${}^{\circ }$/72.78${}^{\circ }$	6.00${}^{\circ }$/63.54${}^{\circ }$
Flex/Ext CF GN	9.69%	0.00${}^{\circ }$/73.58${}^{\circ }$	5.32${}^{\circ }$/63.61${}^{\circ }$
Flex/Ext Kalman GD	5.68%	0.00${}^{\circ }$/70.91${}^{\circ }$	7.36${}^{\circ }$/63.60${}^{\circ }$
Flex/Ext Kalman GN	5.51%	0.00${}^{\circ }$/69.14${}^{\circ }$	6.12${}^{\circ }$/61.71${}^{\circ }$
Flex/Ext Madgwick	5.39%	0.00${}^{\circ }$/65.92${}^{\circ }$	4.48${}^{\circ }$/58.58${}^{\circ }$
Flex/Ext Kinovea	3.79%	2.23${}^{\circ }$/63.82${}^{\circ }$	6.10${}^{\circ }$/63.38${}^{\circ }$

Abbreviations: Flex/Ext = Flexion and Extension; CF = Complementary Filter; GD = Gradient Descending; GN = Gauss-Newton; ROM = range of motion; sd = standard deviation; CI = confidence interval; Var = variance; Min = minimum value found; Max = maximum value found; Min Est = estimated minimum value; Max Est = estimated maximum value.

**Table 3 sensors-21-06477-t003:** Experiment 2—Metrics.

Movement/Filter	ROM Max	Target ROM	Average ROM (sd)
Flex/Ext Madgwick	54.48${}^{\circ }$	–//–	44.62 ± 3.90${}^{\circ }$
Flex/Ext Kinovea	63.06${}^{\circ }$	–//–	50.93 ± 5.96${}^{\circ }$
**Movement/Filter**	**CI 95%**	**Min/Max**	**Min/Max Est**
Flex/Ext Madgwick	7.64%	0.09${}^{\circ }$/54.57${}^{\circ }$	4.82${}^{\circ }$/46.82${}^{\circ }$
Flex/Ext Kinovea	11.67%	3.62${}^{\circ }$/66.69${}^{\circ }$	9.80${}^{\circ }$/60.15${}^{\circ }$

Abbreviations: Flex/Ext = Flexion and Extension; ROM = range of motion; –//– = there was no target; sd = standard deviation; CI = confidence interval; Var = data variance; Min = lowest value found; Max = highest value found; Min Est = minimum value estimate; Max Est = maximum value estimate.

**Table 4 sensors-21-06477-t004:** Experiment 3—Metrics.

Filter	MAPE q${}_{1}$	MAPE q${}_{2}$	MAPE q${}_{3}$	MAPE q${}_{4}$	Average (sd)
CF GD	53.45%	61.24%	68.03%	61.12%	60.96 ± 5.96%
CF GN	50.12%	56.30%	65.18%	56.83%	57.11 ± 6.18%
Kalman GD	52.54%	61.18%	68.25%	62.24%	61.05 ± 6.47%
Kalman GN	51.78%	62.13%	68.34%	60.08%	60.58 ± 6.84%
Madgwick	51.43%	58.97%	65.05%	57.91%	58.34 ± 5.58%
**Filter**	**MAPE q${}_{1}$**	**MAPE q${}_{2}$**	**MAPE q${}_{3}$**	**MAPE q${}_{4}$**	**Average (sd)**
CF GD Abs	9.20%	9.83%	7.53%	7.05%	8.40 ± 1.32%
CF GN Abs	3.02%	3.03%	3.49%	2.27%	2.95 ± 0.51%
Kalman GD Abs	9.95%	12.35%	9.74%	10.63%	10.67 ± 1.18%
Kalman GN Abs	9.63%	11.62%	9.32%	8.84%	9.85 ± 1.22%
Madgwick Abs	4.54%	5.65%	4.37%	3.52%	4.52 ± 0.88%
**Filter**	**RMSE q${}_{1}$**	**RMSE q${}_{2}$**	**RMSE q${}_{3}$**	**RMSE q${}_{4}$**	**Average (sd)**
CF GD	0.69	0.73	0.80	0.71	0.73 ± 0.04
CF GN	0.67	0.73	0.80	0.70	0.73 ± 0.04
Kalman GD	0.66	0.73	0.80	0.71	0.73 ± 0.05
Kalman GN	0.66	0.74	0.80	0.70	0.73 ± 0.05
Madgwick	0.68	0.74	0.79	0.70	0.73 ± 0.04
**Filter**	**RMSE q${}_{1}$**	**RMSE q${}_{2}$**	**RMSE q${}_{3}$**	**RMSE q${}_{4}$**	**Average (sd)**
CF GD Abs	0.14	0.30	0.09	0.10	0.11 ± 0.02
CF GN Abs	0.05	0.04	0.04	0.03	0.04 ± 0.01
Kalman GD Abs	0.11	0.14	0.11	0.12	0.11 ± 0.01
Kalman GN Abs	0.11	0.13	0.10	0.10	0.11 ± 0.01
Madgwick Abs	0.05	0.06	0.05	0.04	0.05 ± 0.01

Abbreviations: q = quaternion; CF = Complementary Filter; GD = Gradient Descent; GN = Gauss-Newton; Abs = Absolute Value. sd = standard deviation; RMSE = Root Mean Squared Error; MAPE = Mean Absolute Percentage Error.

## Data Availability

All codes, notebooks and data used in the work are available on the author’s GitHub (https://github.com/tuliofalmeida/jama (accessed on 1 July 2021) and https://github.com/tuliofalmeida/pyjama (accessed on 1 July 2021)).

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
