# Peer review of "Development of a Low-Cost Open-Source Measurement System for Joint Angle Estimation"

_sensors, 2021, doi:10.3390/s21196477_

Round 1

Reviewer 1 Report

Dear Authors,

This is an interesting manuscript regarding a newly developed low-cost system (HW and SW) for the assessment of joint angles (namely sagittal plane kinematics in this proof of concept) by means of IMU. This is a technical paper with proof-of-concept and should be re-submitted in such format, e.g. as a Communication.

I believe the authors did a great job and proposed a valid idea, which is in line with the trends of limiting the well-known issues of the common motion capture system, i.e., the high costs both in terms of dedicated space, expertise and economic. However, despite the valuable rationale, the authors should focus more on the applicability of their system with regards to the literature and the real-word contexts. I believe that the results obtained are not so promising and accurate as stated by the authors. Many studies have been published with similar scopes, and the authors must explain why their study is actually needed and what adds to the current literature.

Methodology description is quite good and detailed. Some aspects should be moved in/from different sections (see specific comments). The overall architecture of the system is valid. The experiment proposed for the proof-of-concept are correct, but might be improved/enlarged (see specific comments).

Results are coherent with the methodology.

Figures and tables are explicative but can be improved (see specific comments).

Discussion section is a main concern. The authors need to extend this section both regarding their findings and the related literature (see specific comments).

Some typos are present along the manuscript. I suggest a revision from a native speaker.

I encourage the authors to add details to their paper with specific regard to the results obtained and the applicability of the system.

Specific Comments

Title

  • Correct, but I suggest the Manuscript type: Communication

Abstract

  • Should be reshaped to be more fluid. Although the paper has a good level of detail but stays easy to follow, the Abstract section is quite confusing. Some examples: 1. how many sensors were used? 2. How to understand where the MAPE comes from? 3. The Madgwick Filter was the best, but you did not mention you used different filters.
  • Line 1-2: please rephrase to the aim you reported at the end of the introduction section, or use something like "the ... aims to describe a newly developed open... and to provide a proof-of-concept for.."

Introduction

  • Too long. Can be sensibly shortened. Please go straighter to your manuscript’s aim by mentioning relevant literature
  • L15-18: the first sentence is unnecessarily generic. You can merge it with the second one and focus on health systems in motion analysis
  • L19-22: well known, could be shortened as the following on IMU
  • L 23-24: this is a core sentence of your Introduction and should be extended here and above all in your Discussion section. Please focus your written part more on such literature aspects.
  • L43: “we identified”… through a systematic review or simple literature search? Please cite relevant literature. In general, use impersonal language.
  • L47-55: Please move to the Methods section

Methods

  • Overall, well written.
  • L92: which is the maximum distance reachable for a valid transmission?
  • L263: why did you test your device only at 1 km/h? since you had a dedicated robot, you could have assessed your device at multiple speeds. For example, you could have tested your system at the same walking speed of experiment 2 (4/5km/h), so to limit the variables changed among your experiments. This is a main lack of your methodology. I strongly suggest you adding such an assessment to make your analysis more robust.

Results

  • L358-361: please move to the discussion section
  • Apart from the minimum delay, 15° delta in peak angle is a lot. Please address this aspect in the Discussion

Discussion

  • Need a massive improvement. Please discuss your findings more. Report structured comparisons with other devices (low cost systems, maybe). Describe the limitation of your results (e.g., high MAPE, high peak delta) and the reasons why. Describe the limitation of your work methodology.
  • L397-398: please remove
  • L400-401: please move to the methods section
  • L411: good job. I feel this is your highest achievement with this project.
  • L457-459: as commented above, why not testing 4-5 km/h with the Lokomat as well?

Conclusion

  • I feel the errors you find are quite high for real word (or even research) applications, as you also stated. Is there the need for such a system, in its current form?

Tables and Figures

  • Figure 2: relatively easy structure, good job in the light of open systems
  • Table 2/3: The two marco-rows can become one single (e.g., average+-sd; min/Min est, etc) for a clearer read. Also, to me, no need to report the variance.
  • Table 3: 11.67%?

Author Response

Dear reviwer,

Best.

Reviewer 2 Report

In the work, the authors presented a system for measuring joint angles based on inertial sensors. The structure of the device was relatively simple and the method adopted here was facile. The authors conducted some experiments to prove the performance of the system. This manuscript seems to be an experimental report rather than a scientific paper. This work did not provide enough novelty or the importance of scientific experiments. Please consider the following suggestions.

(1) The authors need to clearly point out the innovation and performance superiority of the work relative to similar devices in the article. How far is the performance of the device from the performance of the optimal device, and how will it be improved?

(2) What is the scientific significance of the work?

(3) The authors need to describe in detail how the sensor tests for angle changes.

(4) Why is the measured data so volatile and not smooth? Does this mean that the data during dynamic testing of the device is inaccurate? The authors need to explain it in detail in the manuscript.

(5) The joint stabilization cycle change test at different angles needs to be performed.

Author Response

Dear Reviewer,

Best.

Reviewer 3 Report

  1. The manuscript is concerned with development of low-cost open-source measurement system for joint angles estimation, which is interesting. It is relevant and within the scope of the journal.
  2. However, the manuscript, in its present form, contains several weaknesses. Adequate revisions to the following points should be undertaken in order to justify recommendation for publication.
  3. Full names should be shown for all abbreviations in their first occurrence in texts. For example, JAMA in p.1, 3D in p.3, 2D in p.11, etc.
  4. For readers to quickly catch the contribution in this work, it would be better to highlight major difficulties and challenges, and your original achievements to overcome them, in a clearer way in abstract and introduction.
  5. 1 - JAMA is adopted for data acquisition. What are other feasible alternatives? What are the advantages of adopting this hardware over others in this case? How will this affect the results? The authors should provide more details on this.
  6. 1 - PyJama is adopted for sensor fusion and analysis. What are other feasible alternatives? What are the advantages of adopting this software over others in this case? How will this affect the results? The authors should provide more details on this.
  7. 1 - three experiments are adopted to perform the proof of concept. What are the other feasible alternatives? What are the advantages of adopting these experiments over others in this case? How will this affect the results? More details should be furnished.
  8. 2 - two JAMA’s are adopted to estimate knee joint angle. What are the other feasible alternatives? What are the advantages of adopting these JAMA’s over others in this case? How will this affect the results? More details should be furnished.
  9. 4 - C++ and Python are adopted implement the async-TCP. What are the other feasible alternatives? What are the advantages of adopting these programming languages over others in this case? How will this affect the results? More details should be furnished.
  10. 6 - Complementary Filter with Gradient Descent and Gauss-Newton [17,25], Kalman Filter with Gradient Descent and Gauss-Newton [17,25], and MadgwickAHRS are adopted in the study. What are the other feasible alternatives? What are the advantages of adopting these filters over others in this case? How will this affect the results? More details should be furnished.
  11. 9 - a function presented in Algorithm 4 is adopted to elicit cyclic patterns. What are other feasible alternatives? What are the advantages of adopting this algorithm over others in this case? How will this affect the results? The authors should provide more details on this.
  12. 12 - data from gold-standard devices from the Total Capture dataset [22] is adopted to evaluate the performance of PyJama’s algorithms. What are other feasible alternatives? What are the advantages of adopting this dataset over others in this case? How will this affect the results? The authors should provide more details on this.
  13. 12 - the Mean Absolute Percentage Error metrics is adopted to make all comparisons. What are other feasible alternatives? What are the advantages of adopting this metrics over others in this case? How will this affect the results? The authors should provide more details on this.
  14. 15 - “…the extracted cycles have inverted gait phases - starting from the swing phase to the stance phase. These happened because.…” More justification should be furnished on this issue.
  15. 17 - “…The worst results were the Kalman filters, probably due to the.…” More justification should be furnished on this issue.
  16. 17 - “…the dataset does not provide static body data, and this is probably the explanation why.…” More justification should be furnished on this issue.
  17. 19 - “…presents a flexion peak of 15o in the stance phase and 60o in the swing phase [12,13, 36]. This inversion was probably due to.…” More justification should be furnished on this issue.
  18. The discussion section in the present form is relatively weak and should be strengthened with more details and justifications.
  19. Some key parameters are not mentioned. The rationale on the choice of the particular set of parameters should be explained with more details. Have the authors experimented with other sets of values? What are the sensitivities of these parameters on the results?
  20. Some assumptions are stated in various sections. More justifications should be provided on these assumptions. Evaluation on how they will affect the results should be made.
  21. Moreover, the manuscript could be substantially improved by relying and citing more on recent literatures about real-life applications of soft computing techniques in different fields such as the followings. Discussions about result comparison and/or incorporation of those concepts in your works are encouraged:
  • Banan, A., et al., “Deep learning-based appearance features extraction for automated carp species identification,” Aquacultural Engineering 89: 102053 2020.
  • Shamshirband, S., et al., “A Survey of Deep Learning Techniques: Application in Wind and Solar Energy Resources,” IEEE Access 7 (1): 164650-164666 2019.
  • Fan, Y.J., et al., “Spatiotemporal modeling for nonlinear distributed thermal processes based on KL decomposition, MLP and LSTM network,” IEEE Access 8: 25111-25121 2020.
  1. Some inconsistencies and minor errors that needed attention are: :
  • Replace “…the Madgwick Filterthe showed…” with “…the Madgwick Filter showed…” in line 9 of p.1
  • Replace “…experiment configurations and parameters…” with “…experimental configurations and parameters…” in line 262 of p.11
  • Replace “…Absolute Percentage Error (MAPE) metric…” with “…Absolute Percentage Error (MAPE) metrics…” in line 288 of p.12
  1. In the conclusion section, the limitations of this study and suggested improvements of this work should be highlighted.

Author Response

Dear Reviewer,

Best.

Reviewer 4 Report

The goal of this paper is to propose, develop, and describe an open system for measuring joint angles based on inertial sensors and practical three experiments.

Section 1 should contain an in-depth review of state of the art, but presents after a short introduction some works, lacking a proper comparison.

Figure 1 needs to be improved by adding additional information (the connecting lines are not drawn ok). Perhaps a block diagram is more appropriate in this case. In my view, it does not fit either in the technical or in the typical way of presenting Sensors articles. What are the contributions of the authors and the research innovation brought with this publication?

The authors briefly present the I2C transmission together with the start and stop condition. In addition to Python code, authors need to specify issues related to microcontroller, ISA, design environments, i2C communication, packet loss over TCP.

What operating system is used on ESP32-DevKitC? Because KF works in a real-time loop, it is advisable to present the real-time aspects of the embedded system based on this microcontroller (OS, interrupts, DMA, tasks, real-time responses, i2C communication speed, TCP protocol, GY-80 jitter).  

At what frequency does the microcontroller work and at what communication speed is I2C configured? It's interesting to consider tests that include TCP flow control, connection management, congestion control or traffic constraint in different QoS scenarios. These data can be presented in the form of a table in the "Discussions" section.

A comparison of the results obtained with those presented in section 1 may be included. The authors should clearly clarify the personal contributions for this paper compared to previous research papers. The experimental data presented in the 3 and 4 sections is relatively full.

Author Response

Dear Reviewer,

Best.

Round 2

Reviewer 1 Report

Dear Authors

I appreciate changes you have made in the manuscript.

I think that it could be interesting for our readers in its current form. After revision the paper has consistently improved in quality. I approve of the submission.

Best regards

Reviewer 2 Report

The authors have responded to most of my concerns. I have no other suggestions.

Reviewer 3 Report

The revised paper has addressed all my previous comments, and I suggest to ACCEPT the paper as it is now.

Reviewer 4 Report

The paper was improved by the revision process. I think that the paper can be accepted.